PREREGISTERED RESEARCH ARTICLE

# An empirical appraisal of eLife's assessment vocabulary

**Tom E. Hardwicke** *, **Sarah R. Schiavone**, **Beth Clarke**, **Simine Vazire**

Melbourne School of Psychological Sciences, University of Melbourne, Melbourne, Australia

* tom.hardwicke@unimelb.edu.au

## Abstract

Research articles published by the journal eLife are accompanied by short evaluation statements that use phrases from a prescribed vocabulary to evaluate research on 2 dimensions: importance and strength of support. Intuitively, the prescribed phrases appear to be highly synonymous (e.g., important/valuable, compelling/convincing) and the vocabulary's ordinal structure may not be obvious to readers. We conducted an online repeated-measures experiment to gauge whether the phrases were interpreted as intended. We also tested an alternative vocabulary with (in our view) a less ambiguous structure. A total of 301 participants with a doctoral or graduate degree used a 0% to 100% scale to rate the importance and strength of support of hypothetical studies described using phrases from both vocabularies. For the eLife vocabulary, most participants' implied ranking did not match the intended ranking on both the importance ($n = 59$, 20% matched, 95% confidence interval [15% to 24%]) and strength of support dimensions ($n = 45$, 15% matched [11% to 20%]). By contrast, for the alternative vocabulary, most participants' implied ranking did match the intended ranking on both the importance ($n = 188$, 62% matched [57% to 68%]) and strength of support dimensions ($n = 201$, 67% matched [62% to 72%]). eLife's vocabulary tended to produce less consistent between-person interpretations, though the alternative vocabulary still elicited some overlapping interpretations away from the middle of the scale. We speculate that explicit presentation of a vocabulary's intended ordinal structure could improve interpretation. Overall, these findings suggest that more structured and less ambiguous language can improve communication of research evaluations.

## Introduction

Peer review is usually a black box—readers only know that a research paper eventually surpassed some ill-defined threshold for publication and rarely see the more nuanced evaluations of the reviewers and editor [1]. A minority of journals challenge this convention by making peer review reports publicly available [2]. One such journal, eLife, also accompanies articles with short evaluation statements ("eLife assessments") representing the consensus opinions of editors and peer reviewers [3]. In 2022, eLife stated that these assessments would use phrases drawn from a common vocabulary (Table 1) to convey their judgements on 2 evaluative

---

**Note:** As this is a Preregistered Research Article, the study design and methods were peer-reviewed before data collection. The time to acceptance includes the experimental time taken to perform the study. Learn more about Preregistered Research Articles.

---

**Data Availability Statement:** All data, materials, and analysis scripts are publicly available on the Open Science Framework (https://osf.io/mw2q4/files/osfstorage/). A reproducible version of the manuscript and associated computational environment is available in a Code Ocean container (https://doi.org/10.24433/CO.4128032.v1).

**Funding:** This study was supported by funding awarded to to SV and TEH from the Melbourne School of Psychological Sciences, University of Melbourne. The funders did not play any role in the study design, data collection and analysis, decision to publish, or preparation of the manuscript.

**Competing interests:** SV is a member of the board of directors of The Public Library of Science (PLOS). This role has in no way influenced the outcome or development of this work or the peer-review process, nor does it alter our adherence to PLOS Biology policies on sharing data and materials. All other authors declare they have no conflicts of interest.

**Table 1. Phrases and their definitions (italicised) from the eLife vocabulary representing 2 evaluative dimensions: significance and strength of support.** The significance dimension is represented by 5 phrases and the strength of support dimension is represented by 6 phrases. In a particular eLife assessment, readers only see 1 phrase from each of the evaluative dimensions. Phrases are accompanied by eLife definitions, but these are not shown in eLife assessments (though some words from the definitions may be used).

| eLife vocabulary | |
|---|---|
| **Significance** | **Strength of support** |
| Landmark: *findings with profound implications that are expected to have widespread influence* | Exceptional: *exemplary use of existing approaches that establish new standards for a field* |
| Fundamental: *findings that substantially advance our understanding of major research questions* | Compelling: *evidence that features methods, data, and analyses more rigorous than the current state-of-the-art* |
| Important: *findings that have theoretical or practical implications beyond a single subfield* | Convincing: *appropriate and validated methodology in line with current state-of-the-art* |
| Valuable: *findings that have theoretical or practical implications for a subfield* | Solid: *methods, data, and analyses broadly support the claims with only minor weaknesses* |
| Useful: *findings that have focused importance and scope* | Incomplete: *main claims are only partially supported* |
| | Inadequate: *methods, data, and analyses do not support the primary claims* |

dimensions: (1) "significance"; and (2) "strength of support" (for details see [4]). For example, a study may be described as having "landmark" significance and offering "exceptional" strength of support (for a complete example, see Box 1). The phrases are drawn from "widely used expressions" in prior eLife assessments and the stated goal is to "help convey the views of the editor and the reviewers in a clear and consistent manner" [4]. Here, we report a study which assessed whether the language used in eLife assessments is perceived clearly and consistently by potential readers. We also assessed alternative language that may improve communication.

> **Box 1. A complete example of an eLife assessment. This particular example uses the phrase "important," to convey the study's significance, and the phrase "compelling," to convey the study's strength of support**
>
> "The overarching question of the manuscript is important and the findings inform the patterns and mechanisms of phage-mediated bacterial competition, with implications for microbial evolution and antimicrobial resistance. The strength of the evidence in the manuscript is compelling, with a huge amount of data and very interesting observations. The conclusions are well supported by the data. This manuscript provides a new co-evolutionary perspective on competition between lysogenic and phage-susceptible bacteria that will inform new studies and sharpen our understanding of phage-mediated bacterial co-evolution." [5].

Our understanding (based on [4]) is that eLife intends the common vocabulary to represent different degrees of each evaluative dimension on an ordinal scale (e.g., "landmark" findings are more significant than "fundamental" findings and so forth); however, in our view the intended ordering is sometimes ambiguous or counterintuitive. For example, it does not seem obvious to us that an "important" study is necessarily more significant than a "valuable" study nor does a "compelling" study seem necessarily stronger than a "convincing" study.

Additionally, several phrases like "solid" and "useful," could be broadly interpreted, leading to a mismatch between intended meaning and perceived meaning. The phrases also do not cover the full continuum of measurement and are unbalanced in terms of positive and negative phrases. For example, the "significance" dimension has no negative phrases—the scale endpoints are "landmark" and "useful." We also note that the definitions provided by eLife do not always map onto gradations of the same construct. For example, the eLife definitions of phrases on the significance dimension suggest that the difference between "useful," "valuable," and "important" is a matter of breadth/scope (whether the findings have implications beyond a specific subfield), whereas the difference between "fundamental" and "landmark" is a matter of degree. In short, we are concerned that several aspects of the eLife vocabulary may undermine communication of research evaluations to readers.

In Table 2, we outline an alternative vocabulary that is intended to overcome these potential issues with the eLife vocabulary. Phrases in the alternative vocabulary explicitly state the relevant evaluative dimension (e.g., "support") along with a modifying adjective that unambiguously represents degree (e.g., "very low"). The alternative vocabulary is intended to cover the full continuum of measurement and be balanced in terms of positive and negative phrases. We have also renamed "significance" to "importance" to avoid any confusion with statistical significance. We hope that these features will facilitate alignment of readers' interpretations with the intended interpretations, improving the efficiency and accuracy of communication.

The utility of eLife assessments will depend (in part) on whether readers interpret the common vocabulary in the manner that eLife intends. Mismatches between eLife's intentions and readers' perceptions could lead to inefficient or inaccurate communication. In this study, we empirically evaluated how the eLife vocabulary (Table 1) is interpreted and assessed whether an alternative vocabulary (Table 2) elicited more desirable interpretations. Our goal was not to disparage eLife's progressive efforts, but to make a constructive contribution towards a more transparent and informative peer review process. We hope that a vocabulary with good empirical performance will be more attractive and useful to other journals considering adopting eLife's approach.

Our study is modelled on prior studies that report considerable individual differences in people's interpretation of probabilistic phrases [6–12]. In a prototypical study of this kind, participants are shown a probabilistic statement like "It will probably rain tomorrow" and asked to indicate the likelihood of rain on a scale from 0% to 100%. Analogously, in our study participants read statements describing hypothetical scientific studies using phrases drawn from the eLife vocabulary or the alternative vocabulary and were asked to rate the study's significance/importance or strength of support on a scale from 0 to 100. We used these responses to gauge the extent to which people's interpretations of the vocabulary were consistent with each other and consistent with the intended rank order.

**Table 2. Phrases from the alternative vocabulary representing 2 evaluative dimensions: importance and strength of support. Each dimension is represented by 5 phrases.**

| Alternative vocabulary | |
| --- | --- |
| **Importance** | **Strength of support** |
| Very high importance | Very strong support |
| High importance | Strong support |
| Moderate importance | Moderate support |
| Low importance | Weak support |
| Very low importance | Very weak support |

### Research aims

Our overarching goal was to identify clear language for conveying evaluations of scientific papers. We hope that this will make it easier for other journals/platforms to follow in eLife's footsteps and move towards more transparent and informative peer review.

With this overall goal in mind, we had 3 specific research aims:

- Aim One. To what extent do people share similar interpretations of phrases used to describe scientific research?

- Aim Two. To what extent do people's (implied) ranking of phrases used to describe scientific research align with (a) each other; and (b) with the intended ranking?

- Aim Three. To what extent do different phrases used to describe scientific research elicit overlapping interpretations and do those interpretations imply broad coverage of the underlying measurement scale?

## Methods

Our methods adhered to our preregistered plan (https://doi.org/10.17605/OSF.IO/MKBTP) with one minor deviation: our target sample size was 300, but we accidentally recruited an additional participant, so the actual sample size was 301.

### Ethics

This study was approved by a University of Melbourne ethics board (project ID: 26411).

### Design

We conducted an experiment with a repeated-measures design. Participants were shown short statements that described hypothetical scientific studies in terms of their significance/importance or strength of support using phrases drawn from the eLife vocabulary (Table 1) and from the alternative vocabulary (Table 2). The statements were organised into 4 blocks based on vocabulary and evaluative dimension; specifically, block one: *eLife-significance* (5 statements), block two: *eLife-support* (6 statements), block three: *alternative-importance* (5 statements), block four: *alternative-support* (5 statements). Each participant saw all 21 phrases and responded using a 0% to 100% slider scale to indicate their belief about each hypothetical study's significance/importance or strength of support.

### Materials

There were 21 statements that described hypothetical scientific studies using one of the 21 phrases included in the 2 vocabularies (Table 1). Statements referred either to a study's strength of support (e.g., Fig 1) or a study's significance/importance (e.g., Supplementary Figure A in S1 Text). For the alternative vocabulary, we used the term "importance" rather than "significance." To ensure the statements were grammatically accurate, it was necessary to use slightly different phrasing when communicating significance with the eLife vocabulary ("This is an [phrase] study") compared to communicating importance with the alternative vocabulary ("This study has [phrase] importance"; e.g., Supplementary Figure B in S1 Text).

Additionally, there was 1 attention check statement (Supplementary Figure C in S1 Text), a question asking participants to confirm their highest completed education level (options: Undergraduate degree (BA/BSc/other)/Graduate degree (MA/MSc/MPhil/other)/Doctorate degree (PhD/other)/Other), and a question asking participants the broad subject area of their highest

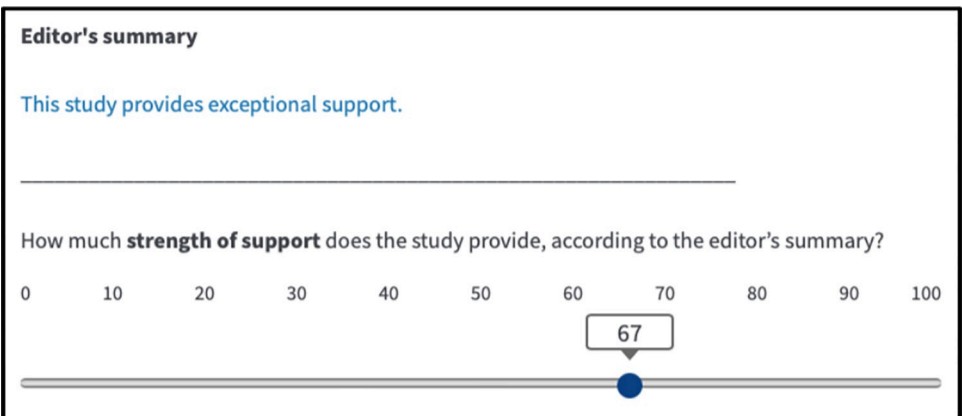

**Fig 1. An example summary statement referring to a study's strength of support and the corresponding response scale with an arbitrary response shown.**

completed education level (options: Arts and Humanities/Life Sciences and Biomedicine/Physical Sciences/Social Sciences/Other). The veridical materials are available at https://osf.io/jpgxe/.

## Sample

**Sample source.** Participants were recruited from the online participant recruitment platform Prolific (https://www.prolific.co/). As of 23rd August 2023, the platform had 123,064 members. Demographic information about Prolific members is provided in S8 Text.

**Sample size.** As data collection unfolded, we intermittently checked how many participants had met the inclusion criteria, aiming to stop data collection when we had eligible data for our target sample size of 300 participants. Ultimately, 461 participants responded to the survey. Of these 461 participants, 156 participants failed the attention check and 12 participants took longer than 30 min to complete the study and were therefore excluded. No participants failed to respond to all 21 statements or completed the study too quickly ($<5$ min). We applied these exclusion criteria one-by-one which removed data from 160 participants and retained eligible data from 301 participants (we unintentionally recruited 1 additional participant).

**Sample size justification.** The target sample size of 300 was based on our resource constraints and expectations about statistical power and precision (see S2 Text).

**Inclusion criteria.** Participants had to have a $\geq 95\%$ approval rate for prior participation on the recruitment platform (Prolific). Additionally, Prolific prescreening questions were used to ensure that the study was only available to participants who reported that they speak fluent English, were aged between 18 and 70 years, and had completed a doctorate degree (PhD/other).

**Procedure.**

1. Data collection and recruitment via the Prolific platform began on September 13, 2023 and was completed on September 14, 2023.

2. After responding to the study advert (https://osf.io/a25vq), participants read an information sheet (https://osf.io/39vay) and provided consent (https://osf.io/xdar7). During this process, they were told that the study seeks to understand "how people perceive words used to describe scientific studies so we can improve communication of research to the general public."

3. Participants completed the task remotely online via the Qualtrics platform. Before starting the main task, they read a set of instructions and responded to a practice statement (S3 Text).

4. For the main task, statements were presented sequentially, and participants responded to them in their own time. The order of presentation was randomized, both between and within the 4 blocks of statements. After each statement, there was a 15-s filler task during which participants were asked to complete as many multiplication problems (e.g., $5 \times 7 =$?) as they could from a list of 10. The multiplication problems were randomly generated every time they appeared using the Qualtrics software. Only numbers between 1 and 15 were used to ensure that most of the problems were relatively straightforward to solve. A single "attention check" statement (Supplementary Figure C in S1 Text) appeared after all 4 blocks had been completed.

5. Participants were required to respond to each statement before they could continue to the next statement. The response slider could be readjusted as desired until the "next" button was pressed, after which participants could not return to or edit prior responses.

6. After responding to all 21 statements and the attention check, participants were shown a debriefing document (https://osf.io/a9gve).

## Results

All analyses adhered to our preregistered plan (https://doi.org/10.17605/OSF.IO/MKBTP). Numbers in square brackets represent 95% confidence intervals computed with the Sison–Glaz method [13] for multinomial proportions or bootstrapped with the percentile method [14] for percentile estimates.

### Participant characteristics

Participants stated that their highest completed education level was either a doctorate degree ($n = 287$) or graduate degree ($n = 14$). Participants reported that the subject areas that most closely represented their degrees were Life Sciences and Biomedicine ($n = 97$), Social Sciences ($n = 77$), Physical Sciences ($n = 57$), Arts and Humanities ($n = 37$), and various "other" disciplines ($n = 33$).

### Response distributions

The distribution of participants' responses to each phrase is shown in Fig 2 (importance/significance dimension) and Fig 3 (strength of support dimension). These "ridgeline" plots [15] are kernel density distributions which represent the relative probability of observing different responses (akin to a smoothed histogram).

Tables 3 and 4 show the 25th, 50th (i.e., median), and 75th percentiles of responses for each phrase (as represented by the black vertical lines in Figs 2 and 3). The tables include 95% confidence intervals only for medians to make them easier to read; however, confidence intervals for all percentile estimates are available in Supplementary Table A in S5 Text and Supplementary Table B in in S5 Text.

### Implied ranking of evaluative phrases

**Do participants' implied rankings match the intended rankings?**   Although participants rated each statement separately on a continuous scale, these responses also imply an overall

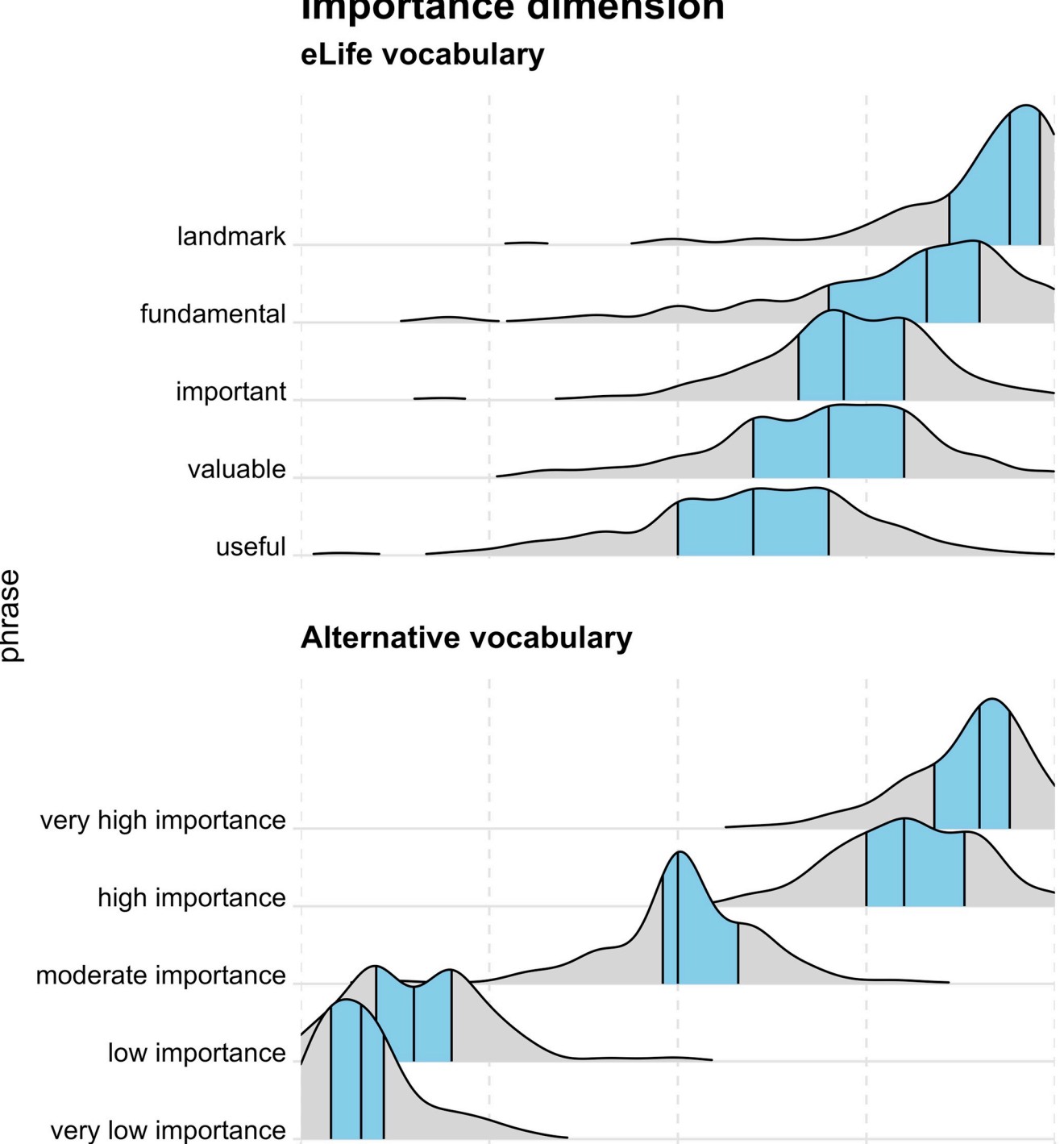

**Fig 2.** Responses to each phrase on the importance/significance dimension as kernel density distributions with the 25th, 50th (i.e., median), and 75th quantiles represented by black vertical lines and the 25th–75th quantile region (i.e., interquartile range) highlighted in blue. The data underlying this figure can be found in https://osf.io/mw2q4/files/osfstorage.

# Strength of support dimension

## eLife vocabulary

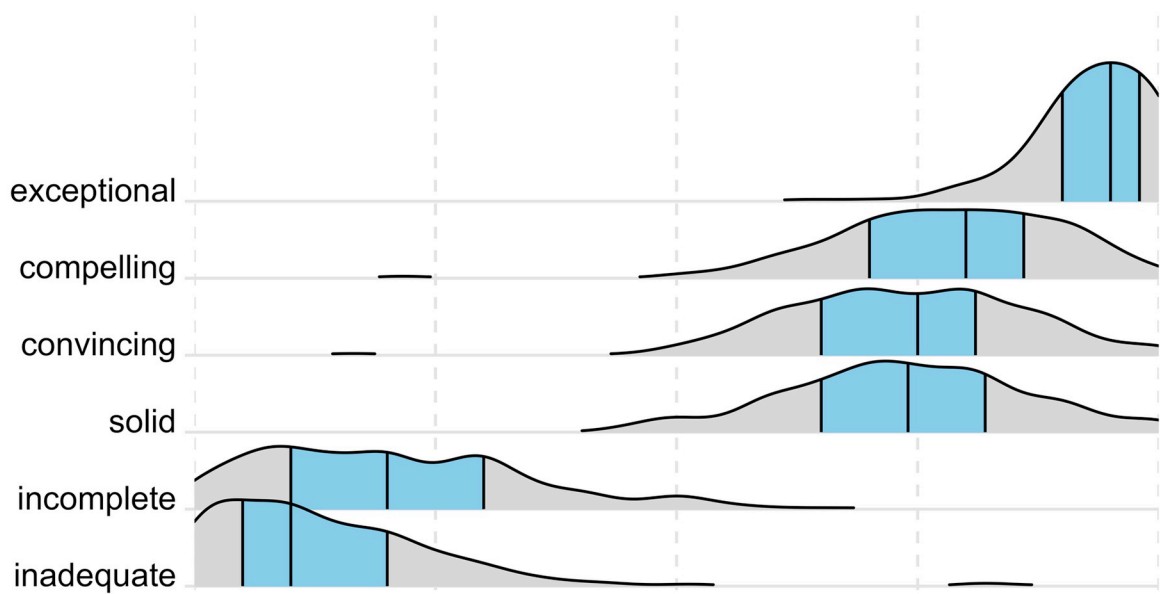

## Alternative vocabulary

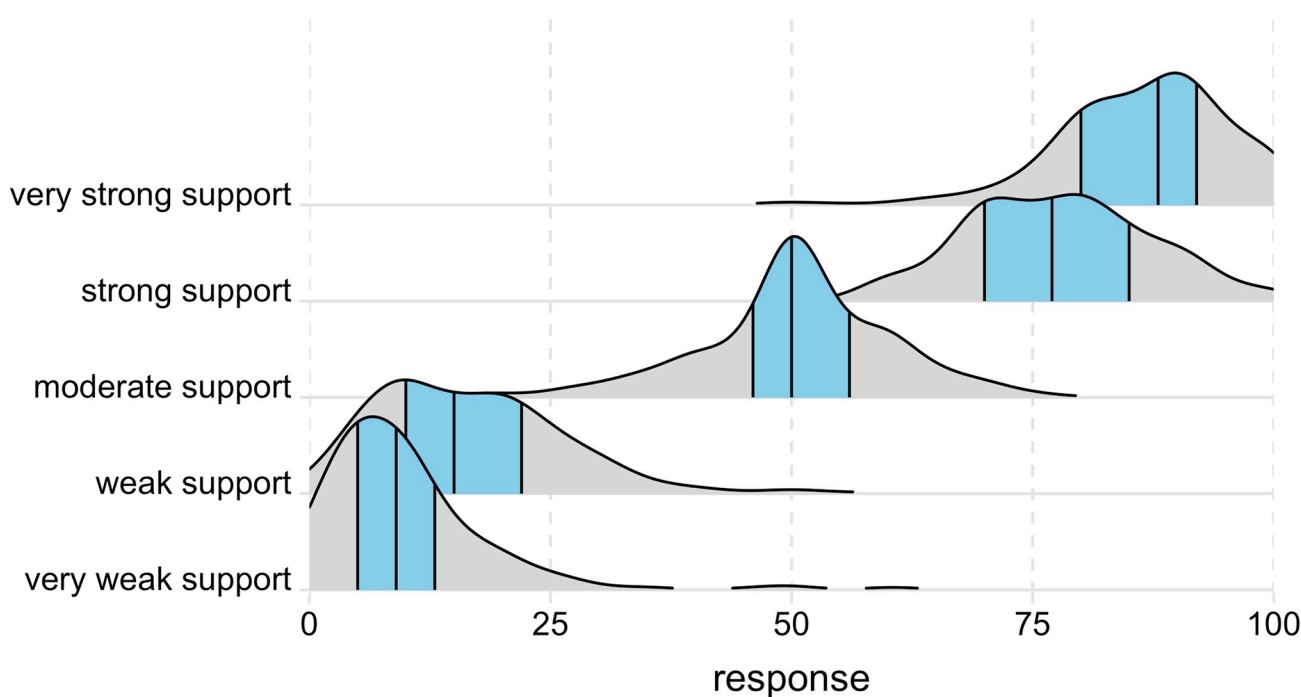

**Fig 3.** Responses to each phrase on the strength of support dimension as kernel density distributions with the 25th, 50th (i.e., median), and 75th quantiles represented by black vertical lines and the 25th–75th quantile region (i.e., interquartile range) highlighted in blue. The data underlying this figure can be found in https://osf.io/mw2q4/files/osfstorage.

**Table 3. Percentile estimates for participant responses to phrases on the significance/importance dimension for eLife and alternative vocabularies.** The data underlying this table can be found in https://osf.io/mw2q4/files/osfstorage.

**Importance dimension**

| Phrase | Percentile | | |
|---|---|---|---|
| | **25th** | **Median [CI]** | **75th** |
| **eLife vocabulary** | | | |
| Landmark | 86 | 94 [92,95] | 98 |
| Fundamental | 70 | 83 [81,85] | 90 |
| Important | 66 | 72 [71,75] | 80 |
| Valuable | 60 | 70 [70,71] | 80 |
| Useful | 50 | 60 [60,63] | 70 |
| **Alternative vocabulary** | | | |
| Very high importance | 84 | 90 [90,91] | 94 |
| High importance | 75 | 80 [80,81] | 88 |
| Moderate importance | 48 | 50 [50,51] | 58 |
| Low importance | 10 | 15 [13,18] | 20 |
| Very low importance | 4 | 8 [6,9] | 11 |

CI, 95% confidence intervals.

ranking of the phrases (in order of significance/importance or strength of support). Ideally, an evaluative vocabulary elicits implied rankings that are both consistent among participants and consistent with the intended ranking. Fig 4 shows the proportion of participants whose implied ranking matched the intended ranking (i.e., "correct ranking") for the different evaluative dimensions and vocabularies.

On the significance/importance dimension, 59 (20% [15% to 24%]) participants' implied rankings of the eLife vocabulary aligned with the intended ranking and 188 (62% [57% to

**Table 4. Percentile estimates for participant responses to phrases on the strength of support dimension for eLife and alternative vocabularies.** The data underlying this table can be found in https://osf.io/mw2q4/files/osfstorage.

**Strength of support dimension**

| Phrase | Percentile | | |
|---|---|---|---|
| | **25th** | **Median [CI]** | **75th** |
| **eLife vocabulary** | | | |
| Exceptional | 90 | 95 [94,95] | 98 |
| Compelling | 70 | 80 [76,80] | 86 |
| Convincing | 65 | 75 [71,75] | 81 |
| Solid | 65 | 74 [71,75] | 82 |
| Incomplete | 10 | 20 [16,20] | 30 |
| Inadequate | 5 | 10 [10,13] | 20 |
| **Alternative vocabulary** | | | |
| Very strong support | 80 | 88 [85,90] | 92 |
| Strong support | 70 | 77 [75,80] | 85 |
| Moderate support | 46 | 50 [50,51] | 56 |
| Weak support | 10 | 15 [14,18] | 22 |
| Very weak support | 5 | 9 [7,10] | 13 |

CI, 95% confidence intervals.

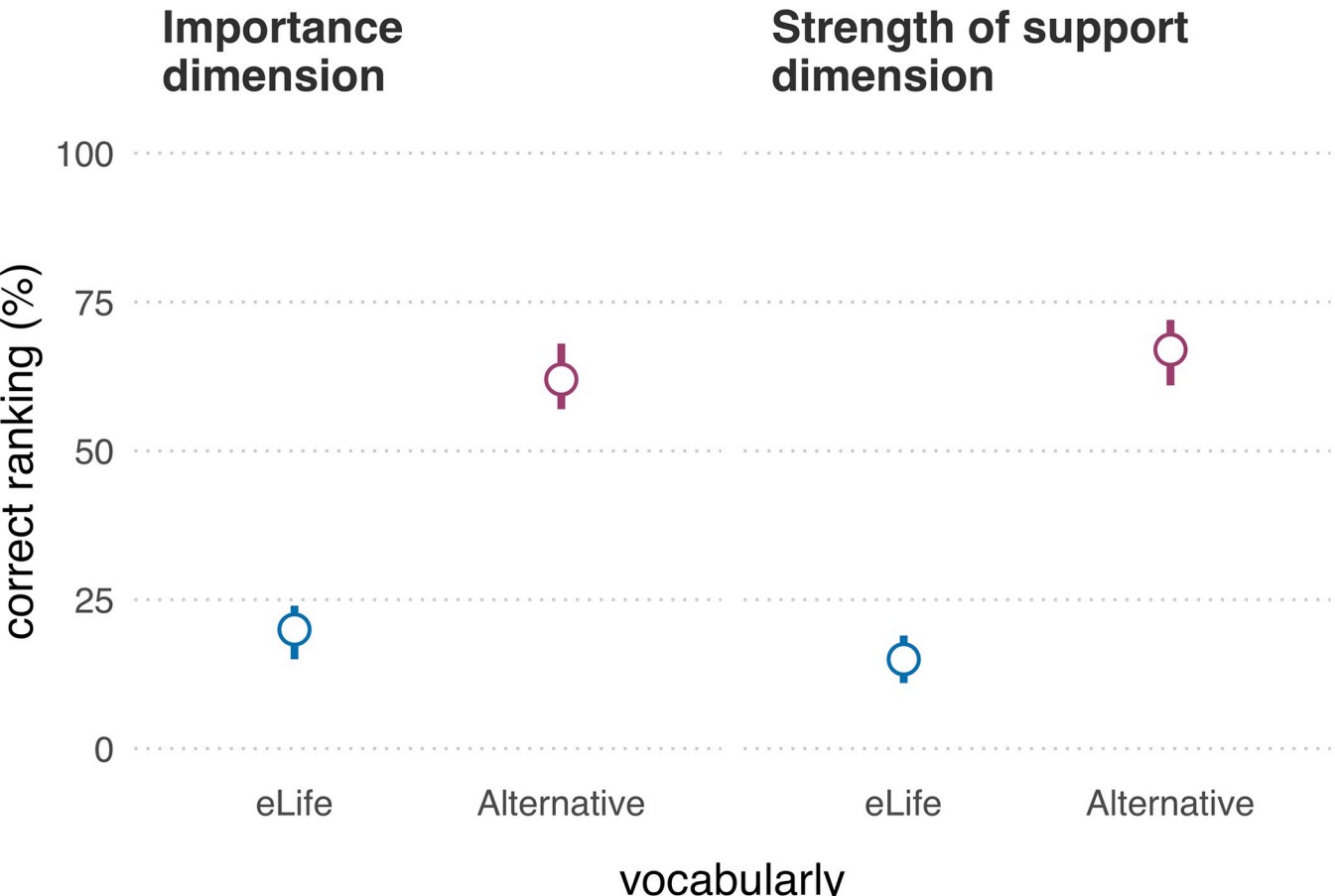

**Fig 4. Proportion of participants (*N* = 301) whose implied ranking matched the intended ranking (i.e., "correct ranking") for both evaluative dimensions and both vocabularies.** Error bars represent 95% confidence intervals using the Sison–Glaz method [13]. The data underlying this figure can be found in https://osf.io/mw2q4/files/osfstorage.

68%]) participants' implied rankings of the alternative vocabulary aligned with the intended ranking. We performed an "exact" McNemar test and computed the McNemar odds ratio with Clopper–Pearson 95% confidence intervals adjusted with the "midp" method, as recommended by [16]. The McNemar test indicated that observing a difference between the vocabularies this large, or larger, is unlikely if the null hypothesis were true (odds ratio = 8.17, 95% CI [5.11, 13.69], *p* = 1.34e-26). The intended ranking was the most popular for both vocabularies; however, participants had 55 different implied rankings for the eLife vocabulary and 8 different implied rankings for the alternative vocabulary (for details, see Supplementary Tables A–D in S6 Text). Note that these values should be compared with caution, as for the significance/importance dimension, the eLife vocabulary had more (6) phrases than the alternative vocabulary (which had 5 phrases and therefore fewer possible rankings).

On the strength of support dimension, 45 (15% [11% to 20%]) participants' ratings of the eLife phrases were in accordance with the intended ranking relative to 201 (67% [62% to 72%]) participants who correctly ranked the alternative vocabulary. A McNemar test indicated that observing a difference between the vocabularies this large, or larger, is unlikely if the null hypothesis were true (odds ratio = 11.4, 95% CI [6.89, 20.01], *p* = 5.73e-35). The intended ranking was the most popular for both vocabularies; though for the eLife vocabulary, an

unintended ranking that swapped the ordinal positions of "convincing" and "solid" came a close second, reflected in the ratings of 44 (15% [10% to 19%]) participants. Overall, there were 34 different implied rankings for the eLife vocabulary, relative to 10 implied rankings for the alternative vocabulary.

**Quantifying ranking similarity.** Thus far, our analyses have emphasised the binary difference between readers' implied rankings and eLife's intended rankings. A complementary analysis quantifies the *degree* of similarity between rankings using Kendall's tau distance ($K_d$) —a metric that describes the difference between 2 lists in terms of the number of adjacent pairwise swaps required to convert one list into the other [17,18]. The larger the distance, the larger the dissimilarity between the 2 lists. $K_d$ ranges from 0 (indicating a complete match) to n(n-1)/2 (where n is the size of one list). Because the eLife strength of support dimension has 6 phrases and all other dimensions have 5 phrases, we report the normalised $K_d$ which ranges from 0 (maximal similarity) to 1 (maximal dissimilarity). Further explanation of $K_d$ is provided in S7 Text.

Fig 5 illustrates the extent to which participants' observed rankings deviated from the intended ranking in terms of normalised $K_d$. This suggests that although deviations from the intended eLife ranking were common, they only tended to be on the order of 1 or 2 discordant rank pairs. By contrast, the alternative vocabulary rarely resulted in any deviations, and when it did, these were typically only in terms of one discordant rank pair.

**Locus of ranking deviations.** So far, we have examined how many participants adhered to the intended ranking (Fig 4) and the extent to which their implied rankings deviated from

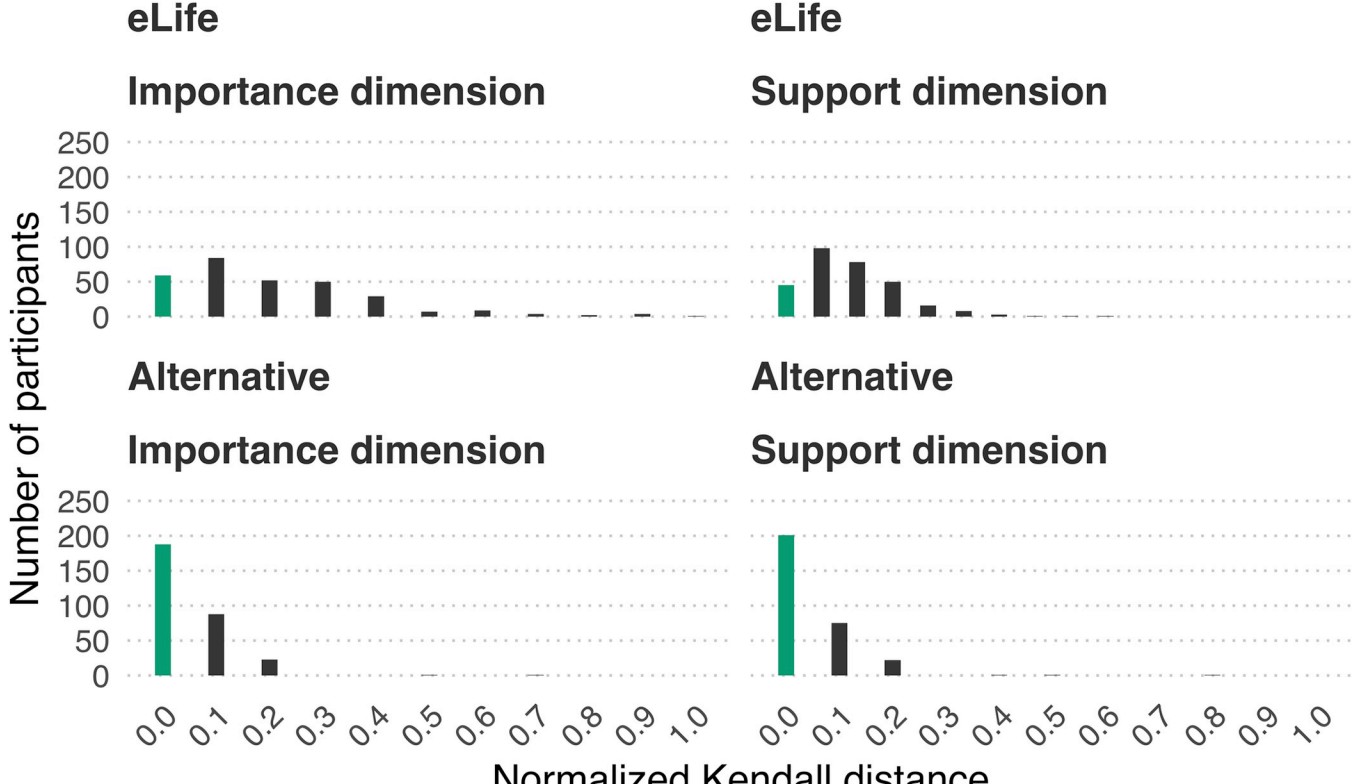

**Fig 5. Extent to which participants (*N* = 301) implied rankings deviated from intended rankings using normalised Kendall's distance.** Zero represents a perfect match (green bar), other values (grey bars) represent increasing dissimilarity from the intended ranking up to 1, which represents maximal dissimilarity. The data underlying this figure can be found in https://osf.io/mw2q4/files/osfstorage.

**eLife vocabulary.
Importance dimension.**

| Observed rank | useful | valuable | important | fundamental | landmark |
|---|---|---|---|---|---|
| landmark | 1 | 4 | 5 | 18 | 72 |
| fundamental | 4 | 16 | 17 | 45 | 18 |
| important | 9 | 31 | 45 | 13 | 3 |
| valuable | 26 | 38 | 23 | 11 | 2 |
| useful | 60 | 12 | 10 | 13 | 4 |

Intended rank

**eLife vocabulary.
Strength of support dimension.**

| Observed rank | inadequate | incomplete | solid | convincing | compelling | exceptional |
|---|---|---|---|---|---|---|
| exceptional | 0 | 0 | 5 | 4 | 8 | 83 |
| compelling | 0 | 0 | 23 | 21 | 44 | 12 |
| convincing | 1 | 0 | 31 | 36 | 27 | 5 |
| solid | 0 | 0 | 40 | 39 | 20 | 0 |
| incomplete | 29 | 69 | 1 | 0 | 0 | 0 |
| inadequate | 69 | 30 | 0 | 0 | 0 | 0 |

Intended rank

**Alternative vocabulary.
Importance dimension.**

| Observed rank | very low importance | low importance | moderate importance | high importance | very high importance |
|---|---|---|---|---|---|
| very high importance | 0 | 0 | 0 | 19 | 80 |
| high importance | 0 | 0 | 1 | 80 | 19 |
| moderate importance | 0 | 1 | 98 | 1 | 0 |
| low importance | 25 | 74 | 1 | 0 | 0 |
| very low importance | 75 | 25 | 0 | 0 | 0 |

Intended rank

**Alternative vocabulary.
Strength of support dimension.**

| Observed rank | very weak support | weak support | moderate support | strong support | very strong support |
|---|---|---|---|---|---|
| very strong support | 0 | 0 | 0 | 15 | 84 |
| strong support | 0 | 0 | 1 | 83 | 15 |
| moderate support | 0 | 1 | 98 | 1 | 0 |
| weak support | 23 | 76 | 0 | 0 | 0 |
| very weak support | 76 | 23 | 0 | 0 | 0 |

Intended rank

Participants (%)
0  25  50  75  100

**Fig 6. Heat maps showing the percentage of participants ($N = 301$) whose implied rankings were concordant or discordant with the intended ranking at the level of individual phrases.** Darker colours and higher percentages indicate greater concordance between the implied rank and the intended rank of a particular phrases. The data underlying this figure can be found in https://osf.io/mw2q4/files/osfstorage.

the intended rankings (Fig 5). However, these approaches do not optimally illustrate *where* the ranking deviations were concentrated (i.e., which phrases were typically being misranked). The heat maps in Fig 6 show the percentage of participants whose implied rankings matched or deviated from the intended ranking at the level of individual phrases. Ideally, a phrase's observed rank will match its intended rank for 100% of participants. For example, the heat

maps show that almost all participants (98%) correctly ranked "moderate importance" and "moderate support" in the alternative vocabulary. The heat maps also reveal phrases that were often misranked with each other, for example: "solid," "convincing," and "compelling" in the eLife vocabulary.

## Discussion

Research articles published in eLife are accompanied by evaluation statements that use phrases from a prescribed vocabulary (Table 1) to describe a study's importance (e.g., "landmark") and strength of support (e.g., "compelling"). If readers, reviewers, and editors interpret the prescribed vocabulary differently to the intended meaning, or inconsistently with each other, it could lead to miscommunication of research evaluations. In this study, we assessed the extent to which people's interpretations of the eLife vocabulary are consistent with each other and consistent with the intended ordinal structure. We also examined whether an alternative vocabulary (Table 2) improved consistency of interpretation.

Overall, the empirical data supported our initial intuitions: while some phrases in the eLife vocabulary were interpreted relatively consistently (e.g., "exceptional" and "landmark"), several phrases elicited broad interpretations that overlapped a great deal with other phrases' interpretation (particularly the phrases "fundamental," "important," and "valuable" on the significance/importance dimension (Fig 2) and "compelling," "convincing," and "useful" on the strength of support dimension (Fig 3)). This suggests these phrases are not ideal for discriminating between studies with different degrees of importance and strength of support. If the same phrases often mean different things to different people, there is a danger of miscommunication between the journal and its readers. Responses on the significance/importance dimension were largely confined to the upper half of the scale, which is unsurprising, given the absence of negative phrases. It is unclear if the exclusion of negative phrases was a deliberate choice on the part of eLife's leadership (because articles with little importance would not be expected to make it through editorial triage) or an oversight. Most participants' implied rankings of the phrases were misaligned with the ranking intended by eLife—20% of participants had aligned rankings on the significance/importance dimension and 15% had aligned rankings on the strength of support dimension (Fig 4). The degree of mismatch was typically in the range of 1 or 2 discordant ranks (Fig 5). Heat maps (Fig 6) highlighted that phrases in the middle of scale (e.g., "solid," "convincing") were most likely to have discordant ranks.

By contrast, phrases in the alternative vocabulary tended to elicit more consistent interpretations across participants and interpretations that had less overlap with other phrases (Figs 2 and 3 and Tables 3 and 4). The alternative vocabulary was more likely to elicit implied rankings that matched the intended ranking—62% of participants had aligned rankings on the significance/importance dimension and 67% had aligned rankings on the strength of support dimension (Fig 4). Mismatched rankings were usually misaligned by one rank (Fig 5). Although the alternative vocabulary had superior performance to the eLife vocabulary, it was nevertheless imperfect. Specifically, interpretation of phrases away from the middle of the scale on both dimensions (e.g., "low importance" and "very low importance") tended to have some moderate overlap (Figs 2, 3, and 6). We do not know what caused this overlap, but, as discussed in the next paragraph, one possibility is that it is overly optimistic to expect peoples' intuitions to align when they judge phrases in isolation, without any knowledge of the underlying scale.

Rather than presenting evaluative phrases in isolation (as occurs for eLife readers and occurred for participants in our study), informing people of the underlying ordinal scale may help to improve communication of evaluative judgements. eLife could refer readers to an external explanation of the vocabulary; however, prior research on interpretation of

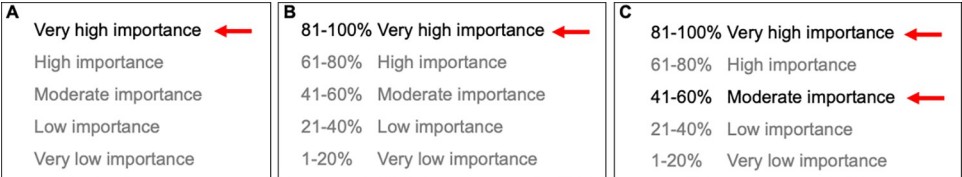

**Fig 7.** Explicit presentation of the intended ordinal structure (A–C), potentially with numerical ranges (B), could improve consistency of interpretation. Judgements by multiple reviewers could also be represented (by different arrows) without forcing consensus (C).

probabilistic phrases suggests this may be insufficient as most people neglect to look up the information [6,19]. A more effective option might be to explicitly present the phrases in their intended ordinal structure [19]. For example, the full importance scale could be attached to each evaluation statement with the relevant phrase selected by reviewers/editors highlighted (Fig 7A). Additionally, phrases could be accompanied by mutually exclusive numerical ranges (Fig 7B); prior research suggests that this can improve consistency of interpretation for probabilistic phrases [19]. It is true that the limits of such ranges are arbitrary, and editors may be concerned that using numbers masks vague subjective evaluations in a veil of objectivity and precision. To some extent we share these concerns; however, the goal here is not to develop an "objective" measurement of research quality, but to have practical guidelines that improve accuracy of communication. Specifying a numerical range may help to calibrate the interpretations of evaluators and readers so that the uncertainty can be accurately conveyed. Future research could also explore the relationship between the number of items included in the vocabulary and the level of precision that reviewers/editors wish to communicate.

Our study has several important limitations. First, we did not address whether editor/reviewer opinions provide valid assessments of studies or whether the vocabularies provide valid measurements of those opinions. We also note that eLife assessments are formed via consensus, rather than representing the opinions of individuals, which raises questions about how social dynamics may affect the evaluation outcomes. It may be more informative to solicit and report individual assessments from each peer reviewer and editor, rather than force a consensus (e.g., see Fig 7C). Although these are important issues, they are beyond the scope of this study, which is focused on clarity of communication.

Second, we are particularly interested in how the readership of eLife interpret the vocabularies, but because we do not have any demographic information about the readership, we do not know the extent to which our sample is similar to that population. We anticipated that the most relevant demographic characteristics were education status (because the content is technical), knowledge of subject area (because eLife publishes biomedical and life sciences), and language (because the content is in English). All of our participants reported speaking fluent English, the vast majority had doctoral degrees, and about one third had a degree in the Biomedical and Life Sciences. Relative to this sample, we expect the eLife readership probably consists of more professional scientists, but otherwise we think the sample is likely to be a good match to the target population. Also note that eLife explicitly states that eLife assessments are intended to be accessible to non-expert readers [4], therefore, our sample is still a relevant audience, even if it might contain fewer professional scientists than eLife's readership.

Third, to maintain experimental control, we presented participants with very short statements that differed only in terms of the phrases we wished to evaluate. In practice however, these phrases will be embedded in a paragraph of text (e.g., Box 1) which may also contain "aspects" of the vocabulary definitions (Table 1) "when appropriate" [4]. It is unclear if the

inclusion of text from the intended phrase definitions will help to disambiguate the phrases and future research could explore this.

Fourth, participants were asked to respond to phrases with a point estimate; however, it is likely that a range of plausible values would more accurately reflect their interpretations [9,11]. Because asking participants to respond with a range (rather than a point estimate) creates technical and practical challenges in data collection and analysis, we opted to obtain point estimates only.

## Conclusion

Overall, our study suggests that using more structured and less ambiguous language can improve communication of research evaluations. Relative to the eLife vocabulary, participants' interpretations of our alternative vocabulary were more likely to align with each other, and with the intended interpretation. Nevertheless, some phrases in the alternatively vocabulary were not always interpreted as we intended, possibly because participants were not completely aware of the vocabulary's underlying ordinal scale. Future research, in addition to finding optimal words to evaluate research, could attempt to improve interpretation by finding optimal ways to present them.

## Supporting information

**S1 Text. Example stimuli and attention check.**
(DOCX)

**S2 Text. Sample size planning.**
(DOCX)

**S3 Text. Task instructions.**
(DOCX)

**S4 Text. Peer Community in Registered Reports Design Template.**
(DOCX)

**S5 Text. Expanded versions of Table 3 and Table 4 percentile estimates with confidence intervals.**
(DOCX)

**S6 Text. Frequency of implied rankings.**
(DOCX)

**S7 Text. Further explanation of Kendall's distance.**
(DOCX)

**S8 Text. Demographics of Prolific members.**
(DOCX)

## Acknowledgments

The Stage 1 version of this preregistered research article has been peer-reviewed and recommended by Peer Community in Registered Reports (https://rr.peercommunityin.org/articles/rec?id=488).

## Research transparency statement

The research question, methods, and analysis plan were peer reviewed and preregistered as a Stage One Registered Report via the *Peer Community in Registered Reports* platform (https://doi.org/10.17605/OSF.IO/MKBTP). There was only 1 minor deviation from the preregistered

protocol (target sample size = 300; actual sample size = 301). A *Peer Community in Registered Reports* design table is available in S4 Text. All data, materials, and analysis scripts are publicly available on the Open Science Framework (https://osf.io/mw2q4/files/osfstorage/). A reproducible version of the manuscript and associated computational environment is available in a Code Ocean container (https://doi.org/10.24433/CO.4128032.v1).

## Author Contributions

**Conceptualization:** Tom E. Hardwicke, Simine Vazire.

**Data curation:** Tom E. Hardwicke.

**Formal analysis:** Tom E. Hardwicke.

**Funding acquisition:** Tom E. Hardwicke.

**Investigation:** Tom E. Hardwicke, Sarah R. Schiavone, Beth Clarke, Simine Vazire.

**Methodology:** Tom E. Hardwicke.

**Project administration:** Tom E. Hardwicke.

**Resources:** Tom E. Hardwicke.

**Software:** Tom E. Hardwicke.

**Supervision:** Tom E. Hardwicke, Simine Vazire.

**Validation:** Tom E. Hardwicke.

**Visualization:** Tom E. Hardwicke.

**Writing – review & editing:** Tom E. Hardwicke, Sarah R. Schiavone, Beth Clarke, Simine Vazire.

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
