## [Editor Report · Decision Letter 0]

25 Apr 2024

Dear Tom, 

Thank you for submitting your ex-PCI-RR manuscript entitled "Finding the right words to evaluate research: An empirical appraisal of eLife’s assessment vocabulary" for consideration as a Meta-Research Article by PLOS Biology.

Your stage 2 Pre-Registered Research Article manuscript has now been evaluated by the PLOS Biology editorial staff, as well as by an academic editor with relevant expertise, and I'm writing to let you know that we would like to send your submission out for re-review.

IMPORTANT: Please can you change the article type to "Pre-Registered Research Article" when you upload your additional metadata (see next paragraph)?

Once your full submission is complete, your paper will undergo a series of checks in preparation for re-review. After your manuscript has passed the checks it will be sent out for review. To provide the metadata for your submission, please Login to Editorial Manager (https://www.editorialmanager.com/pbiology) within two working days, i.e. by Apr 29 2024 11:59PM.

Kind regards,

Roli

Roland Roberts, PhD

Senior Editor

PLOS Biology

rroberts@plos.org

---

## [Decision Letter · Decision Letter 1]

21 May 2024

Dear Dr Hardwicke,

Thank you for your patience while your revised registered report "Finding the right words to evaluate research: An empirical appraisal of eLife’s assessment vocabulary" went through peer-review at PLOS Biology. I am sending you this decision letter on behalf of my colleague Roli Roberts, who is out of the office this week, in order to avoid unnecessary delays. Your manuscript has now been evaluated by the PLOS Biology editors, an Academic Editor with relevant expertise, and by the three independent reviewers that assessed the stage 1 submission at PCI-RR. All of the reviewers have signed their reports: Reviewer #1 is Ross Mounce, Reviewer #2 is Štěpán Bahník and Reviewer #3 is Veli-Matti Karhulahti.

In light of the reviews, which you will find at the end of this email, we are pleased to offer you the opportunity to address the following points, which we anticipate should not take you very long. We will then assess your revised manuscript and your response to the reviewers' comments with our Academic Editor without further rounds of peer-review.

1) Please add a column to the right of the study design table that reports the actual outcome per row and whether it confirmed or disconfirmed the hypothesis.

2) As you will see, both R1 and R3 remain uncomfortable about your level of engagement with eLife, uncomfortable enough to raise this independently, because they did not have a chance to see a revised stage 1. Although it is not a precondition for publication, we would tend to agree and encourage you to engage with eLife about their intention behind the assessment system, as this would make the discussion more straightforward and certain (eg about intentionality). In addition, this work is likely to receive a fair amount of scrutiny post-publication, including the peer-review file, and so the reviewer qualms might be discussed in post-publication commentary around the article.

3) Reviewer 2 also speaks of additional analyses that could be done with the data. Again, this would not be required for publication, but we did want to give you the opportunity to address this report if you wanted. Especially the assessment of whether the frequency of actual ratings might be justified given pre-selection based on quality, or whether the reviewers disproportionately use words on the lower end of the scale, appears to be relatively doable in a reasonable time frame, should you wish to complement the work in that way.

4) Could the COI statement for author Simine Vazire be modified in our submission system to state the following? 

"SV is a member of the board of directors of The Public Library of Science (PLOS). This role has in no way influenced the outcome or development of this work or the peer-review process, nor does it alter our adherence to PLOS Biology policies on sharing data and materials."

**IMPORTANT - SUBMITTING YOUR REVISION**

Please submit the following files along with your revised manuscript:

1. A 'Response to Reviewers' file - this should detail your responses to the editorial requests, present a point-by-point response to all of the reviewers' comments, and indicate the changes made to the manuscript (if any).

*Resubmission Checklist*

*Published Peer Review*

Please note while forming your response, if your article is accepted, you will have the opportunity to make the peer review history publicly available. For the sake of transparency, we strongly recommend that you opt into this. The record will include editor decision letters (with reviews) and your responses to reviewer comments. If eligible, we will contact you to opt in or out. Please see here for more details:

*PLOS Data Policy*

Sincerely,

Nonia

Nonia Pariente, PhD

Editor in Chief

PLOS Biology

npariente@plos.org

on behalf of

Roland

Roland Roberts, PhD

Senior Editor

PLOS Biology

rroberts@plos.org

REVIEWS:

Reviewer #1: Ross Mounce

Reviewer #2: Štěpán Bahník

Reviewer #3: Veli-Matti Karhulahti

Reviewer #1: I re-read the stage 1 protocol submission, which is registered here: https://doi.org/10.17605/OSF.IO/MKBTP (which I will hereafter refer to simply as S1)

The rationale and the stated hypotheses of this Stage 2 manuscript submitted to PLOS Biology (which I will hereafter refer to simply as S2), are indeed the same as approved in S1. The wording between the S1 and S2 is not precisely the same. 

A brief digression / sidenote: not a fault of the authors, but I must remark it is a little unhelpful to provide reviewers only PDF files when asking peer reviewers to check the similarity between two separate documents with separate formatting (one with line numbering, one without!). I converted the S1 and S2 into plain text files, chopping-off the S2 at the appropriate places to ensure maximum comparability with the content of the S1. From here a simple bag of words analysis comparing between the plain text of S1 and S2 revealed to me a few linguistic changes that seem to have occurred between S1 and S2 e.g.

In S1: "eLife recently stated that these assessments will use phrases drawn from a common vocabulary…"

In S2: "eLife stated that these assessments would use phrases drawn from a common vocabulary…" 

Although there are small changes of wording detectable between the S1 and S2 I did not perceive them to have changed the meaning of the introduction, rationale, or stated hypotheses.

I note that the authors aimed for a target sample size of 300 in the stage 1 protocol and report a target sample size of 301 in the stage 2 manuscript. This tiny difference is noted by the authors in the stage 2 manuscript and is NOT a significant deviation from the stage 1 protocol.

As far as I can tell the authors have adhered to the approved S1 experimental procedures.

The data are to an extent able to test the authors proposed hypotheses. 

As I noted in my review of S1 I still think it is a bit of a cop-out by the authors to choose not to ask eLife for the broad (e.g. country-level) demographic characteristics of eLife readers - most well-run journals absolutely do hold data on the geography of where in-bound page views are coming from, at least to the country-level if not in much finer detail via standard web traffic analysis captured by e.g. Google Analytics or Matomo. I can see from inspection of their HTML that eLife use Google Analytics to track website usage data. Their global site tag is currently "GTM-WVM8KG" - you'll see it tagged in every single page. They will know the IP address of every visitor to their site and from the IP address will be able to determine the geolocation of the reader with a high degree of accuracy (relatively few readers take steps to obfuscate the IP address they are reading from). 

The conclusions of S2 are valid given the data. However, the conclusions do pre-suppose that eLife intend their vocabulary to have a clear and unambiguous ranking order between words. What if the apparent ambiguity that the data & research has uncovered was _intentional_ on eLife's part? It's not clear to me if the authors here have ever asked eLife if they intended all the words in the vocabulary to have a strict and clear order of ranking. Might the authors want to ask eLife about this? If the intention was to have a clear and unambiguous ranking, why wouldn't eLife have just used numbers rather than words (which clearly as this study and many others show, can be interpreted differently by different people)? Some of the wording in the S2 rather assumes a certain intentionality on behalf of eLife and I'm not 100% convinced that the authors here can speak confidently of eLife's intentionality without directly asking eLife.

I have examined the data deposited at OSF (https://doi.org/10.17605/OSF.IO/MW2Q4) to support S2. The data is in a public repository, it is appropriately described, rights-waived under CC0 1.0 to enable maximum re-use potential, and it is time-stamped (April 6, 2024) to demonstrate that the data was collected after the S1 approval. The data complies with the PLOS data availability policy.

Conclusion: Whilst I retain and reiterate my quibbles from the S1 review stage, this S2 does represent sound, pre-registered research and should be published.

Reviewer #2: The second stage of the registered report follows the plan from the first stage. I did not notice any deviations. The results are clearly described and the text in discussion and conclusion is justified given the results.

My biggest regret for the study is that it only purports to show that the eLife vocabulary is not optimal, that the alternative vocabulary is better, but it does not purports to show that the alternative vocabulary is good itself. This is correctly described in the conclusion, where the authors note that future research could find optimal vocabulary, so I do not think the description needs to be changed. I just hope that the alternative vocabulary will not be adopted based only on the results of this study. As the authors correctly note, the wording of the alternative vocabulary overlaps (at the two most extreme points on each end of the scale), which suggests that a different wording might be better. An informative and useful vocabulary should take into account not only whether the descriptions are correctly ordered, but also how well they cover the importance and support in actual studies. This would be possible to quantify, but the authors opted not do that during the first stage of the registered report. Relatedly, the authors note in the discussion that the eLife vocabulary covers only the upper half of the scale. It would be possible to look at the frequency of actual ratings to see whether this might be justified given pre-selection based on quality, or whether the reviewers disproportionately use words on the lower end of the scale.

Reviewer #3: Thank you for inviting me to review the Stage 2 of this manuscript. I was excited about the plan at Stage 1 and it's indeed interesting to read the results at Stage 2. First, as far as my assessment goes, the authors have followed the plan carefully, the materials and data are shared, the solitary hypotheses was successfully tested, and the paper thus deserves to be accepted as well as further debated in post-publication discussion. For editors, I acknowledge to have reviewed the html script but didn't run the code in R and am (still) not qualified to comment on the details of Kendall's tau distance.

When reading the final version of the paper in reflection with the Stage 1 review, I must re-address a few points that were raised earlier. Because there was no re-assessment for the revised Stage 1 plan after revisions, I did not have an earlier opportunity to comment on the final Stage 1. Therefore, my Stage 2 review revisits some issues of Stage 1; this is mainly for the open review record and may not entail action from authors. To be clear, these points were considered resolved by the editors/recommenders, so the authors should not be demanded to resolve them but optionally engage if and as they best see fit.

1. One of my and other reviewers' comments at Stage 1 was to further involve eLife in the study. Considering that the paper systematically uses words like "intended" as the core epistemic lens through which the data are analysed, it still feels awkward that "intentions" have not been backed up beyond the journal's public statement "…to help convey the views of the editor and the reviewers in a clear and consistent manner." Reading author responses post hoc, I can see there has been an informal dinner with one editor before the study, but this was only to inform them about the study. Of course it could be that the short description on eLife website represents all there is to the intended purpose, but that remains unknown for readers at least. I'd suggest having a representative of eLife review the full Stage 2 in order to ensure that their intentions have not been misinterpreted. E.g., it feels strange to publish sentences like "It is unclear if the exclusion of negative phrases was a deliberate choice on the part of eLife's leadership" when it all it takes to figure that out is ask. Moreover, it would've been valuable to hear what specifically eLife means by "consistency" that is operationalised in a specific way here.

2. Related to this, while we still don't know what eLife fully intended, their stated decision to use "common vocabulary" implies that the idea was indeed not to use overly quantifying ranking (Likert scale agreement, numeric scores, etc.) but in their own words "rich" vocabulary that perhaps serves to balance between ranking and the inherent vagueness of scientific assessment where comparisons between studies rarely make sense. One could reasonably argue that the results of the study support such balance: eLife vocabulary appears to be somewhat successful at communicating correctly ordered rankings, while not expressing overly confidence on the quantifiability of such ranking. In this regard, it's possible that e.g. reporting percentages (proposed on p. 24) would move the assessment system even more far away from eLife's intended epistemology and/or values. The authors are aware of this, but due to lacking involvement of eLife, all this remains speculation for readers (which could be easily cleared). 

3. As commented at Stage 1, the paper has one explicit registered test for a hypothesis, which remains difficult for readers to separate from the rest of analytic context on page 17: "The McNemar test indicated that observing a difference between the vocabularies this large, or larger, is unlikely if the null hypothesis were true (odds ratio = 8.17, 95% CI [5.11,13.69], p = 1.34e-26)." The authors must be commende

---

## [Editor Report · Decision Letter 2]

27 Jun 2024

Dear Tom,

Thank you for your patience while we considered your revised manuscript "Finding the right words to evaluate research: An empirical appraisal of eLife’s assessment vocabulary" for publication as a Preregistered Research Article at PLOS Biology. This revised version of your manuscript has been evaluated by the PLOS Biology editors and the Academic Editor.

Based on our Academic Editor's assessment of your revision, we are likely to accept this manuscript for publication, provided you satisfactorily address the following data and other policy-related requests.

IMPORTANT - please attend to the following:

a) Please remove the first phrase from the title, making it simply "An empirical appraisal of eLife’s assessment vocabulary."

b) Thank you for adding the extra column to the design table (currently Table D1 in the supplement); please could you move this Table to the main paper and adjust the manuscript to accommodate it?

c) Please can you include the following rubric in an "Acknowledgements" section (after the "Conclusion"): "The Stage 1 version of this pre-registered research article has been peer-reviewed and recommended by Peer Community In Registered Reports (https://rr.peercommunityin.org/articles/rec?id=488)". [This is a formula that we have used previously in manuscripts that have come to us down the PCI route]

d) Thanks for providing the underlying data in OSF. Please cite the location of the data clearly in all relevant main Figure legends, e.g. “The data underlying this Figure can be found in S1 Data” or “The data underlying this Figure can be found in https://osf.io/mw2q4/files/osfstorage"

e) Please move your ethics statement [“This study was approved by a University of Melbourne ethics board (project ID: 26411).”] to an appropriate part of the Methods section.

f) Please remove the funding information and competing interests statement from the manuscript file; it seems that these are already in the Editorial Manager metadata, from which they will automatically be pulled.

g) Please move the "Research transparency statement" into the main manuscript. Perhaps next to the Acknowledgements? (I don't have a precedent for this, so one of my Production colleagues might move it later)

We expect to receive your revised manuscript within two weeks. 

*Published Peer Review History*

*Press*

Sincerely,

Roli

Roland Roberts, PhD

Senior Editor

rroberts@plos.org

PLOS Biology

DATA NOT SHOWN?

---

## [Editor Report · Decision Letter 3]

9 Jul 2024

Dear Tom,

Thank you for the submission of your revised Preregistered Research Article "An empirical appraisal of eLife’s assessment vocabulary" for publication in PLOS Biology. On behalf of my colleagues and the Academic Editor, Christopher Chambers, I'm pleased to say that we can in principle accept your manuscript for publication, provided you address any remaining formatting and reporting issues. These will be detailed in an email you should receive within 2-3 business days from our colleagues in the journal operations team; no action is required from you until then. Please note that we will not be able to formally accept your manuscript and schedule it for publication until you have completed any requested changes.

Sincerely, 

Roli

Senior Editor

PLOS Biology

rroberts@plos.org